# Anthrax Vaccines in the 21st Century

**DOI:** 10.3390/vaccines12020159

**Published:** 2024-02-03

**Authors:** Apostolos P. Georgopoulos, Lisa M. James

**Affiliations:** 1The Gulf War Illness Working Group, Brain Sciences Center, Department of Veterans Affairs Health Care System, Minneapolis, MN 55417, USA; lmjames@umn.edu; 2Department of Neuroscience, University of Minnesota Medical School, Minneapolis, MN 55455, USA; 3Department of Psychiatry, University of Minnesota Medical School, Minneapolis, MN 55455, USA; 4Department of Neurology, University of Minnesota Medical School, Minneapolis, MN 55455, USA

**Keywords:** anthrax, *Bacillus anthracis*, vaccination, protective antigen, chronic multisymptom illness (CMI), gulf war illness (GWI)

## Abstract

Vaccination against *Bacillus anthracis* is the best preventive measure against the development of deadly anthrax disease in the event of exposure to anthrax either as a bioweapon or in its naturally occurring form. Anthrax vaccines, however, have historically been plagued with controversy, particularly related to their safety. Fortunately, recent improvements in anthrax vaccines have been shown to confer protection with reduced short-term safety concerns, although questions about long-term safety remain. Here, we (a) review recent and ongoing advances in anthrax vaccine development, (b) emphasize the need for thorough characterization of current (and future) vaccines, (c) bring to focus the importance of host immunogenetics as the ultimate determinant of successful antibody production and protection, and (d) discuss the need for the systematic, active, and targeted monitoring of vaccine recipients for possible Chronic Multisymptom Illness (CMI).

## 1. Introduction

Anthrax is a rare but deadly disease in humans, against which vaccination is the best preventive measure. Given the potential use of aerosolized anthrax spores as a bioweapon, and after the mass vaccinations of thousands of troops during the first Persian Gulf War of 1990–1991, a large effort has been expended to improve anthrax vaccines with respect to their content, effectiveness, tolerance, administration, adverse effects, and storage. More specifically, the aim is to produce anthrax vaccines that will be well characterized as to their content, highly protective pre- and post-exposure, minimally reactogenic, administered in a few doses, without long-term adverse effects, and safe for special populations (e.g., children, pregnant or lactating women, older people, immunocompromised individuals, etc.). In addition, they would be produced relatively quickly and retain potency for long periods of storage. Although these properties may look daunting, such a project is feasible, in general, assuming, of course, that a firm commitment and sufficient funding would be made available. In this review, we survey the field from a bird’s eye view regarding the anthrax vaccine’s checkered past, quiet present, and hopeful future.

## 2. Anthrax Toxins

There are three anthrax toxin-related proteins: Protective Antigen (PA), Lethal Factor (LF), and Edema Factor (EF). These proteins are harmless in isolation by themselves, but their combinations inside the cell yield the following deadly anthrax toxins: Lethal Toxin (LT; PA + LF), Edema Toxin (ET; PA + EF), and the anthrax toxin (LT, ET). The Protective Antigen is an 83 kDa protein (PA83), called so because it is used in vaccines to protect from the disease. PA83 binds to receptors (ANTXR1 and ANTXR2) at the cell surface and is cleaved by furin (a cellular proteolytic enzyme) to a smaller protein, PA63 (63 kDa). Clusters of PA63 molecules (PA63 oligomers, typically heptamers and octamers) bind 3–4 molecules of LF and EF, and the complex enters the cell through clathrin-mediated endocytosis. In the acid environment of the cytosol, LT and ET are formed, causing inflammation and apoptosis. Ultimately, LT and ET cause the death of the host by damaging the cardiovascular system and the liver, respectively [1].

## 3. Anthrax Vaccines

### 3.1. Antigens

All anthrax vaccines contain antigenic material of protein(s) of *Bacillus anthracis* (*B. anthracis).* Different formulations of anthrax vaccines are licensed for use in humans in different countries [2]. In the United States (USA), the Anthrax Vaccine Adsorbed (AVA; BioThrax^®^) [3] is prepared from “cell-free filtrates of microaerophilic cultures of an avirulent, nonencapsulated strain of *B. anthracis*. The final product, prepared from the sterile filtrate culture fluid, contains proteins, including the 83 kDa protective antigen protein. The final product contains no dead or live bacteria. One dose (0.5 mL) is formulated to contain anthrax antigen filtrate: 50 micrograms (50 µg) adsorbed on aluminum hydroxide (0.6 mg aluminum per dose)” [3]. Aluminum is added as an adjuvant to enhance immunogenicity. More recently, the oligodeoxynucleotide compound CPG 7909 has been used as an additional adjuvant to AVA and has been licensed in the USA for post-exposure prophylaxis under the brand name CYFENDUS^TM^ [4], administered subcutaneously. The strain of *B. anthracis* used to produce the current AVA vaccine is not available. Originally, in 1963, the “Strain V770-NPl-R, a nonencapsulated, nonproteolytic, and avirulent mutant of *B. anthracis*, was used” [5]; whether the same strain is used for the production of the AVA BioThrax^®^ vaccine is not known.

In the United Kingdom (UK), the Anthrax Vaccine Precipitated (AVP) is an alum precipitate of a sterile culture filtrate of the *B. anthracis* Sterne (34F2) strain. Its contents have been recently characterized precisely and were found to contain “at least 138 *B. anthracis* proteins, including PA (65%), Lethal Factor (8%) and Edema Factor (3%), using liquid chromatography–tandem mass spectrometry.” [6].

In Russia, a Live Attenuated Anthrax Vaccine (LAAV, or LAV for short) is used [7], which, unlike AVA and AVP, contains live spores of the attenuated *B. anthracis* STI strain. The history of the development of this vaccine and its extensive testing in field studies on thousands of people are described in [8]. Similarly, a live attenuated vaccine has been in use in China [9]. Details of the AVA, AVP, and LAV vaccines are given in Table 1 of [2].

AVA and AVP are given intramuscularly at multiple doses, initially over a period of months, followed by annual boosts. LAV is administered as a two-dose regimen via scarification, subcutaneous injection, or in aerosolized form, followed by annual boosters. Concise reviews of anthrax vaccine development and challenges can be found in [10,11].

### 3.2. Adjuvants

Vaccines are typically administered with adjuvants to enhance their immunogenicity, induce local inflammatory reactions, and overall promote the production of antibodies against the vaccine antigen. A commonly used adjuvant is aluminum (alum) [12], which is the adjuvant in AVA and AVP vaccines. CYFENDUS [4] is AVA with the additional adjuvant CpG 7909 [13,14]. Addavax, a squalene oil-in-water adjuvant, has also been tested successfully [15]. In a different effort to enhance immunogenicity, BA3338, a surface layer homology domain-possessing protein, has been found to enhance immune response when added to a PA + alum-based vaccine [16] and similarly encapsulating PA within trimethyl chitosan nanoparticles [17]. A comprehensive review of the different kinds of adjuvants used in studies of anthrax vaccines can be found in [18].

## 4. Antibodies to Anthrax Proteins and Anthrax Toxins

All four vaccines (AVA, AVP, LAV, and CYFENDUS) induce the production of antibodies against anthrax proteins but with different profiles [19,20,21]. In a study comparing antibody production in humans induced by these vaccines [20], it was found that AVA and LAV induced the production only of anti-PA antibodies, whereas AVP induced the production of anti-PA, anti-LF, and anti-EF antibodies. However, the titers of anti-PA antibodies were higher in the AVA and LAV vaccines than in the AVP vaccine. Antibodies against anthrax spores were also detected following LAV administration [7].

Anthrax vaccines lead to the production of anthrax Toxin-Neutralizing Activity (TNA) [22,23,24], for which anti-PA antibodies seem to be the main contributors [22], as well as anti-LF antibodies [24]. From a regulatory perspective, TNA has been used for AVA licensure, namely a TNA NF50 (50% neutralization factor) level of 0.56 in rabbits and 0.29 in nonhuman primates [3].

Anti-PA antibody titers are commonly assumed to parallel conferred protection against disease upon exposure and are thus taken as a proxy for effective vaccine protection. However, this is a gray zone. First, levels of anti-PA antibodies do not ensure commensurate protection [25]. And second, although antibody titers can be determined in both humans and experimental animals (mice, rats, guinea pigs, rabbits, and monkeys) [26], the degree of protection can only be determined in experimental animals, where there is substantial heterogeneity in congruence between presence of antibodies and degree of protection among different species [25,27]. Finally, different levels of protection after the same vaccination have been observed between male and female mice [25].

Given the aforementioned qualifications, it is generally agreed upon that the TNA is the common standard for evaluating the performance of a particular vaccine at a particular time following immunization: the higher the TNA, the better the quality of regarded vaccine’s performance. In that context, it is reassuring that there is a highly positive correlation between anti-PA IgG titers and TNA activity [28]. Although this holds overall for combined, aggregate data, there is a differentiation among IgG classes and among the PA domains against which the antibodies are directed [29]. More specifically, it was found that (a) anti-PA antibodies with a high IgG4/IgG1 ratio are less potent on Lethal Toxin-Neutralizing Activity (LTNA), and (b) IgG1 antibodies directed against epitopes from PA domain 4 are more potent on LTNA [29]. These factors would naturally affect the ultimate protection of the vaccinee against anthrax exposure. (a) With respect to IgG subclass distribution, it was found in a recent real-world study of 144 AVA vaccinees [29] that almost half of the cohort (47.9%) neutralized LTNA poorly, and almost a third (30.6%) neutralized LTNA very poorly; poor neutralizers had an IgG4/IgG1 > 1.5 ratio. This finding has an important implication in that IgG4 levels have been found to increase with more AVA doses [28], implying a potential gradual loss of protection over time under the standard booster schedule. (b) With respect to targeted anti-PA domains by the AVA vaccine, it was found that vaccinees with IgG-targeting PA domain 4 neutralized LTNA much better than those with antibodies targeting other domains. The identification of these two independent factors influencing LTNA (IgG4/IgG1 ratio and percent of IgG-targeting PA domain 4) is an important step in understanding the intricate factors ultimately conferring protection against anthrax exposure. Practically, with respect to the IgG4/IgG1 ratio, the goal would be to find a vaccination schedule that would keep this ratio low. On the other hand, with respect to the PA domain effect, the goal would be to develop PA-subunit vaccines using domain 4 epitopes, as discussed below.

## 5. Newer Vaccines

Although current vaccines are considered effective, their substantial reactogenicity and “undefined composition, lot-to-lot variation, and extensive dosing regimen” (page D29 in [10]) have driven efforts for the development of new-generation anthrax vaccines. To that end, substantial effort has been expended to (a) develop precisely characterized vaccines, (b) explore different compositions of vaccines in addition to PA alone, and (c) develop vaccines inducing quick and long-lasting antibody production, thus minimizing the number of initial dosages and later boosts, (d) minimize reactogenicity (local reactions and short-term systemic adverse effects), and (e) improve conditions for long storage of the vaccine without loss of potency.

### 5.1. Vaccines with Killed but Metabolically Active (KBMA) B. anthracis

A whole-bacterial-cell anthrax vaccine utilizing killed but metabolically active *B. anthracis* that has been developed [30]. The Sterne strain was genetically engineered (a) to render it photochemically sensitive so as to be inactivated while remaining metabolically active, and (b) to allow the secretion of inactive but immunogenic PA, LF, and EF. This vaccine was avirulent in mice and less reactogenic on injection than alum-adsorbed recombinant PA (rPA). In addition, it offered excellent protection against anthrax challenge and elicited the production of antibodies against numerous anthrax antigens, including high levels of anti-PA and toxin-neutralizing antibodies. The main advantage of this vaccine is that it contains various other antigens, in addition to PA/LF/EF, and thus offers the chance for elicitation of antibodies against a variety of antigens with a broad coverage against various strains of *B. anthracis.*

### 5.2. Vaccines with rPA, PA Subunits, and Chimeric LF-PA Combinations

As mentioned above, AVA and AVP are derived from the supernatant of cultures of acellular, nonvirulent strains of *B. anthracis*. AVA contains PA, whereas AVP also contains small amounts of LF and EF [6]. The lack of precise composition and the presence of lot-to-lot variation has led to the development of recombinant PA (rPA) vaccines, which use PA synthesized by various organisms (e.g., *E. coli. B., anthracis., B. subtilis.* and others), or in a plant-based system [31]. These vaccines typically have a shorter shelf life than the ones currently in use, and efforts are directed toward increasing their storage life using various approaches [18]. These vaccines present a significant advance over the current ones with respect to well-defined PA content and reduced (or lack) of lot-to-lot variation. In addition, they have been shown to elicit good titers of PA-neutralizing antibodies, confer protection against challenge with anthrax, and be well tolerated and without major side effects [32].

There has been substantial research on identifying PA-subunits with high immunogenicity as vaccine candidates [29,33,34]. Antibodies targeting PA domain 4 possess high TNA [29,33], thus qualifying well for that purpose. Such “rationally designed vaccines” [33], composed of PA subunits inducing antibodies with high TNA, would be expected to be optimally effective in anthrax protection and superior to whole-PA-based vaccines (e.g., AVA), which induce antibodies that target, to a large extent, the 20 kDa PA20 region of the PA83 molecule, i.e., the inactive part of PA83, which is separated from it after cleavage by furin-like membrane proteases [35,36]. An epitope-based vaccine comprising the D1 LF domain and PA domain 4 was found to be very protective [37]. Similarly, a chimeric vaccine containing the N-terminal region of LF and segments from PA induced the formation of antibodies recognizing PA, LF, and EF, which was similarly effective to a simple PA vaccine and also more stable [38]; a different formulation of an LF-PA vaccine was found to elicit a robust humoral response in a mouse model [39]. Finally, a triple-chimeric vaccine has been designed using the N-terminal domains of LF and EF and the C-terminal domain of PA, tested in guinea pigs with strong humoral results [40].

### 5.3. Plasmid-Based Vaccines

Anthrax plasmid DNA vaccines encoding genetically detoxified PA and LF were found to be immunogenic and protect rabbits against challenge with aerosolized anthrax spores [41]. Adenovirus-based PA vaccines have been shown to be effective in producing good immune responses and protection from challenge [42]. A vaccine consisting of adenovirus vector Ad5-encoding PA and administered intranasally in a single dose has been reported to confer partial protection from inhalation anthrax in mice [43]. Similarly, an Ad5-coded PA vaccine administered intramuscularly induced rapid and robust immune responses and effective protection from challenge in rats [44]. Finally, recombinant anthrax vaccines containing chimeric fusion of components of PA with LF [45] and vaccines with attenuated spores of mutated anthrax Sterne strain have also been developed [46].

### 5.4. mRNA-Based Vaccines?

The advent and success of mRNA-based vaccines for COVID-19 pave the way for the possible development of mRNA-based vaccines for anthrax. For now, it is not clear if that is feasible or, if so, what the advantages/drawbacks would be of such an anthrax vaccine. Nevertheless, the potential is there to be explored as needed.

## 6. Effective Protection from Anthrax in Humans: The Critical Role of Host Immunogenetics (Human Leukocyte Antigen; HLA)

### 6.1. The Dependence on Host Immunogenetics

All vaccines above are considered effective in protecting from anthrax. Given the seriousness of the disease, the degree of protection is tested by exposing experimental animals to anthrax and recording their survival. Several animal species have been used for that purpose, including mice, rats, guinea pigs, rabbits, and monkeys. In general, vaccine protection is similar across species but also with substantial variation [25]. A field study in humans recorded a case of anthrax in a subject fully vaccinated with the AVA vaccine [47]. In fact, effective protection of vaccinated people upon exposure depends critically on the production of suitable antibodies against anthrax toxins and/or spores (see below). The production of such antibodies depends, in turn, on the genetic immune makeup (HLA Class II genes) of the vaccinated subject [6,48] and their general immunocompetence; the actual degree of protection conferred by the vaccine would be the result of both the vaccination and the host immune system, as discussed in more detail below.

### 6.2. The Vaccine–HLA Match Challenge

As mentioned above, the real-life efficacy of a given vaccine in humans would depend on the ability of the vaccinee to make antibodies against the vaccine peptide(s), a complicated process which, when successful, begins with the formation of a peptide–HLA molecule complex and ends with the production of antibodies against the bound antigen by B cells, via transport of the complex from the antigen-presenting cell (APC) to B cells by CD4+ lymphocytes [49]. Any interference with this process would impair antibody production, as is the case, for example, in various diseases, such as total absence of lymphocytes, reduction in lymphocyte numbers (e.g., due to radiation), immunodeficiency due to AIDS, drug-induced immunosuppression, etc. However, the first and most crucial step is the successful formation of a stable complex between a vaccine peptide and an HLA Class II molecule [49]. A protein contained in the vaccine (e.g., PA) is fragmented by proteases in APC cells to ~15-*mer* to 22-*mer* fragments, some of which bind with high affinity to one or more of six classical HLA Class II molecules in the APC cell. These six alleles come from three genes (two from each DPB1, DQB1, and DRB1 gene), which means that a good match is not always guaranteed. The “goodness” of the match depends on the binding affinity between the peptide and the HLA molecule; in the best case (where there is an excellent match of binding of various antigen peptides to all six HLA Class II molecules), the chance of making antibodies is very high, but in the worst case of absence of high affinity binding for any of the six HLA Class II molecules a person carries, the chance of making antibodies is meager. The fact that good antibody responses have been frequently observed for the same vaccine (e.g., containing PA) means that there is sufficient variation among the six HLA Class II molecules to bind with high affinity to PA peptide fragments (epitopes). 

## 7. Safety

### 7.1. Local and Systemic Adverse Events

Naturally, a constant focus has been on the safety of anthrax vaccines. Unlike other vaccines, the anthrax vaccine is highly reactogenic with appreciable local (redness, induration, edema, itching, or tenderness) and systemic (fever, malaise, and myalgia) symptoms [50]. According to the World Health Organization [51], mild local adverse events such as erythema and/or edema, pruritus, and induration at the site of injection are very common following both AVA and AVP vaccines. Systemic adverse events are also commonly reported following both vaccines, including myalgia, rash, headache, malaise, joint pain, nausea, vomiting, loss of appetite, chills, and fever. Severe adverse events reported after AVA administration include allergic or inflammatory reactions at the injection site and anaphylaxis. Infrequently reported single events included disorders of the nervous system, skin, subcutaneous tissue, musculoskeletal system, connective tissue, and bones. (Specific references regarding the adverse events above can be found in [51]).

### 7.2. Past Issues with AVA Manufacturing

Safety concerns were amplified for two main reasons. First, there were repeated problems in the US regarding the AVA vaccine’s standards of manufacturing by the various companies in the decade of 1990s and beginning of the 2000s, problems with conforming to regulations of the Food and Drug Administration (FDA) and recommendations [52,53]. This means that the vaccine during the 1990–1991 Gulf War and afterward was under a cloud. Although its use was strictly speaking “legal”, covered by several interim authorizations by the FDA pending the implementation by the manufacturing company of FDA recommendations, it was not until January 2001 that the FDA completed an approval and licensure process, allowing the manufacturing plant to resume production of the anthrax vaccine [52]. After a comprehensive review of these issues and of symptoms and complaints of armed services personnel and veterans of the 1990–1991 Gulf War, the General Accounting Office (GAO) Report made the following recommendation: “We recommend that the Secretary of Defense direct the establishment of an active surveillance program (unlike the passive VAERS) to identify and monitor adverse events associated with each anthrax vaccine immunization. This program should ensure that appropriate and complete treatment and follow-up are provided to those who have experienced adverse events and to those who may experience them in the future” (page 24 in [52]). Unfortunately, these recommendations were rejected by the Department of Defense for reasons that were not found adequate by the GAO Committee (pages 24–26 in [52]).

### 7.3. Chronic Multisystem Illness (CMI)

#### 7.3.1. Gulf War Illness (GWI)/CMI

The second reason for raising safety concerns about the anthrax vaccine (AVA and AVP) was the emergence of multisystem complaints from many Gulf War veterans in the US and UK shortly after their enlisting in the Gulf War and extending during the decade of 1990s. The disorder was termed Gulf War Illness [54,55,56,57]. The Center for Disease Control (CDC) case definition [55] requires one or more symptoms from at least two of the following categories (duration ≥ 6 months): (1) fatigue; (2) mood and cognition (symptoms of feeling depressed, difficulty in remembering or concentrating, feeling moody, feeling anxious, trouble in finding words, or difficulty in sleeping); and (3) musculoskeletal (symptoms of joint pain, joint stiffness, or muscle pain). This symptom constellation is now called Chronic Multisymptom Illness (CMI) ([57,58]).

GWI/CMI is a serious, debilitating disease independent of other disorders, such as Post-Traumatic Stress Disorder (PTSD). It has been observed in about 30% of veterans of the 1990–1991 Gulf War, those who were both deployed in the field as well as those who were nondeployed, with higher rates in the former. Detailed studies during the past 20-odd years have documented abnormalities and/or dysfunction in multiple organ systems, including brain atrophy [59], immune dysfunction [60,61], chronic inflammatory state [62,63,64,65], and autoimmunity [66,67], among others, as well as an overlap with known autoimmune disorders such as rheumatoid arthritis, multiple sclerosis, and Sjögren’s syndrome [67,68].

Originally, GWI was attributed to chemical exposures, including nerve gas, burning pits, insecticides, etc. [69]. Although these could be contributing factors, two facts point clearly to other basic causes, these facts being (a) the occurrence of the disease in nondeployed veterans (thus excluding exposure to burning pits and insecticides) and (b) the occurrence of CMI (using the CDC case definition criteria above) in veterans returning from service in Operation Iraqi Freedom/Operation Enduring Freedom (OIF/OEF), where there were no nerve gas exposures [70]. More specifically, mild-to-moderate CMI was diagnosed in 49.5% of veterans at 1 y after returning from service, and severe CMI was present in 10.8% of veterans [70]; CMI could not be accounted for by PTSD or predeployment, potentially predisposing conditions [70]. Remarkably, the CMI rates above are higher than the ~30% rate of occurrence of CMI/GWI following the 1990–1991 Gulf War. These data point to other factors at the root of this disorder.

#### 7.3.2. GWI/CMI, HLA and Anthrax Vaccine

In 2016, we reported the discovery of six HLA Class II alleles that protected veterans from GWI (DPB1*01:01, DPB1*06:01, DQB1:08:02, DRB1*01:01, DRB1:08:11, and DRB1:13:02) [71]. These alleles not only discriminated 84.1% correctly GWI veterans from healthy control but also, more importantly, the average severity of GWI symptoms decreased linearly with the average number of protective alleles (Figure 2 in [71]). In addition, the presence of the DRB1*13:02 allele above prevented brain atrophy in GW veterans who carried it [72], as well as other brain abnormalities [73]. These results point to a biological factor as a contributing cause of CMI/GWI. Such a factor could be the anthrax vaccine, which was administered to the US (AVA) and the UK and associated countries (e.g., Canada, Australia) (AVP). Interestingly, two of the protective HLA alleles above (DRB1*01:01 and DRB1*13:02) possess high estimated binding affinities to PA [74] and other proteins contained in the AVP vaccine [29], lending support to the hypothesis that the protection they conferred against GWI [71] could be mediated by the successful production of antibodies against PA and other proteins.

Additional evidence for the possible association of GWI and anthrax vaccination was provided recently by the finding that the serum of GWI patients induces apoptosis (cell death) in neural cultures in vitro [75] and that this detrimental effect is ameliorated by the addition of anti-PA antibody (polyclonal [76] or monoclonal [77]) in the culture (Figure 1) [77]. This finding documents the contribution of PA (or a PA fragment targeted by the anti-PA antibodies) to the neurotoxicity exerted by the GWI serum. The neurotoxicity of PA alone was confirmed, and the diverse mechanisms by which it is exerted were further investigated in a subsequent study [77].

### 7.4. Testing for Anthrax Vaccine Safety

Unfortunately, the focus on short-term effects and gross assessment of health and organ anatomy has carried through in testing newer versions of the anthrax vaccine [14]. The delay in appearance of GWI symptoms (typically months to a year) and their nature (fatigue, muscle pain, neurocognitive symptoms, etc.) necessitate the assessment of possible anthrax vaccine involvement by more sophisticated testing over longer periods of time than current assessment protocols [14]. For example, vaccinated animals should be assessed for the presence of fatigue, muscle/joint pain, neurological signs, etc., and performance in suitable (to the animal) behavioral/cognitive tasks one year after vaccination, when CMI symptoms were documented in veterans [70]. With respect to humans, the assessment for CMI, the hallmark of possible anthrax vaccination, should be carried out routinely and annually for all vaccinated veterans, irrespective of deployment. The CMI questionnaire is easily administered, is based on self-report, and can be filled electronically.

### 7.5. Development of New Anthrax Vaccines → Road to Safety

The safety issue of the anthrax is an ongoing concern. Thousands and thousands of armed forces personnel are being routinely administered the anthrax vaccine in a compulsory fashion in the US, and yet the exact composition of the AVA vaccine they are receiving is not known. In a pioneer study mentioned above, Modi et al. [6] fully characterized the AVP vaccine and found that it contained 138 proteins, some of them in abundance. Until that time, AVP was thought to contain PA and traces of LF—a significant contrast to the results of the full AVP characterization [6]. The safety profile of those proteins is unknown, e.g., their potential to trigger autoimmunity via molecular mimicry, epitope expansion, tapping cryptic epitopes, etc. Also unknown is their potential for chronic inflammation, a classic substrate for autoimmunity. Obviously, “safety” is not referred to only as a sentinel event of life or death but also as an illness. And a substantial percentage of military personnel (and veterans thereof) are suffering from Chronic Multisymptom Illness (CMI) [55,56,70]. Now, the development of future anthrax vaccines, as discussed above, is very important, not only for immunogenicity, storage, etc. reasons, but, to us, mostly for safety reasons. Given the unknown exact composition of AVA and the startling revelation of the multi-protein composition of AVP, recombinant vaccines are a big step forward because their composition and purity are exactly known. Even recombinant PA vaccines are an improvement, and even better, recombinant subunit (especially LF-PA chimeric) vaccines that contain selected immunogenic portions of the anthrax toxin proteins, as discussed above. Therefore, the development of new vaccines is inextricably linked to improving the safety of such vaccines in the future.

## 8. Closing Remarks and Recommendations

The following are our key take-home messages and recommendations.

(a) All anthrax vaccines need to be thoroughly characterized in the same way that the AVP vaccine was [6]. Traditionally, it was stated that AVP contains “protective antigen PA and the lethal factor” (page 145 in [2]); yet the study of Modi et al. [6] identified 138 proteins in the vaccine, of which 8 were “abundant”, including PX01 (PX01-90), AD (Alcohol Dehydrogenase), Chap. (60 kDa Chaperonin), Eno (Enolase), and PGK (Phosphoglycerate Kinase), in addition to PA, LF, and LT. This characterization is especially important to be performed for AVA since it is currently being administered routinely to a large number of armed forces personnel.

(b) Rationally designed, epitope-based, and LF-PA chimeric vaccines are excellent prospects for the production of well-characterized anthrax vaccines inducing antibodies with high toxin-neutralizing activity and high level of protection.

(c) Host immunogenetics, i.e., the HLA Class II genetic makeup of a vaccinee, is the ultimate determinant for effective protection expected to be offered by a vaccine. In silico investigation of binding affinities of vaccine epitopes and specific HLA Class II alleles would be very helpful in predicting vaccine effectiveness for particular individuals or populations, as demonstrated recently in the case of SARS-CoV2 vaccines [78].

(d) As reviewed above, the history and severity of CMI/GWI with multi-organ involvement and immune dysfunction; autoimmunity; its presence in nondeployed personnel; the protection conferred by specific HLA Class II alleles (e.g., DRB1*01:01 and DRB1*13:02) with high estimated binding affinities to PA and other anthrax vaccine (AVP) proteins; the gradual and continuous deterioration of the general health of US Gulf War veterans; and, last but not least, the prima facie evidence for the presence of neurotoxicity in the serum of veterans afflicted with GWI, neurotoxicity that can be partially inhibited/reversed by specific anti-PA antibody; this constellation of diverse evidence linking GWI and anthrax vaccination strongly speaks for the need to active monitoring of anthrax vaccinees, e.g., by evaluating annually the wellness of cardinal CMI symptoms (fatigue, pain, neurocognitive) that have been documented in a large percentage of veterans [70], with a follow-up evaluation of specific organ systems (e.g., brain), if needed. The issue at stake, namely the health of military vaccinees, is serious enough to warrant the establishment of routine active health assessment and monitoring beyond the common passive reporting. Comprehensive reviews of CMI during the past 30-odd years attest to this need [79,80,81,82,83,84].

## Figures and Tables

**Figure 1 vaccines-12-00159-f001:**
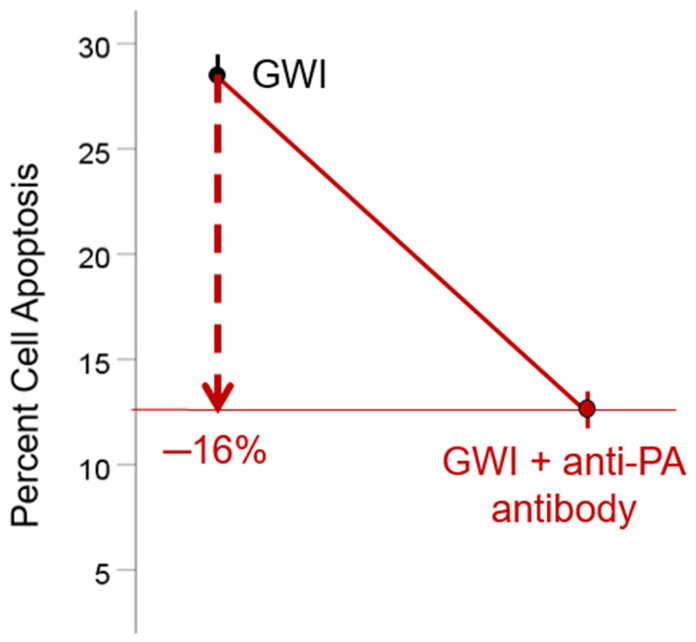
Beneficial effect of anti-PA antibody in reducing neurotoxicity of GWI serum. Apoptosis (cell death) of cultured N2A cells induced by GWI serum is reduced by 16% (*p* < 0.001) by the addition of anti-PA antibody. Apoptosis in controls was 7.8%. Data shown are adapted from Figures 7 and 8 in ref [77].

## Data Availability

Not applicable.

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
