# Peer review of "Anthrax Vaccines in the 21st Century"

_vaccines, 2024, doi:10.3390/vaccines12020159_

Round 1

Reviewer 1 Report

Comments and Suggestions for Authors

This review is very well written and sections 1-5 are very good.

Major criticisms:  

1)  '6. Safety'  The second paragraph assumes and essentially asserts that the anthrax vaccine is the cause of GWI and because of this, the anthrax vaccine will need "rigorous assessment of various organ systems over months and years is necessary".  Such assessments will never be accomplished in a modern vaccine trial and it is ironic that this review touts rushed mRNA vaccine technology - the exact opposite of this point.

2)  '7. Neurotoxicity...'  This entire paragraph should be reduced to 2-3 sentences and folded into the previous section or section 9.  Readers looking for a review on the anthrax vaccine likely have little interest in the in vitro neurotoxicity of PA63.

3) '8.3. PA-free vaccines'  I don't really understand this paragraph at all, with regard to 15-aa epitopes.  This paragraph should be about subunit vaccines, much work has gone into the feasibility of a domain 4 vaccine.  Various possibilities of using PA without PA-20 and domain 3 and PA-domain/LF chimeras should be discussed.  Furthermore, the assertion that vaccine trials need to determine the vaccine's potential for inducing autoimmunity is untenable.  What would those experiments even be?  It would take decades for proper human data. 

4) '9. Persistence of PA in the blood' A paragraph needs to be added about the immunologic features of AVA and persistent exposure to the vaccinating antigen due to slow release and frequent boosts.  Several studies have shown that the AVA schedule causes induction of IgG4 subclass anti-PA which may have a detrimental effect on neutralization capacity.  The recent ACIP update has somewhat addressed this (citations below).  This is very applicable to this review.

Quinn CP, Sabourin CL, Schiffer JM, Niemuth NA, Semenova VA, Li H, Rudge TL, Brys AM, Mittler RS, Ibegbu CC, Wrammert J, Ahmed R, Parker SD, Babcock J, Keitel W, Poland GA, Keyserling HL, El Sahly H, Jacobson RM, Marano N, Plikaytis BD, Wright JG. Humoral and Cell-Mediated Immune Responses to Alternate Booster Schedules of Anthrax Vaccine Adsorbed in Humans. Clin Vaccine Immunol. 2016 Apr 4;23(4):326-38. doi: 10.1128/CVI.00696-15. PMID: 26865594; PMCID: PMC4820509.

Smith K, Garman L, Norris K, Muther J, Duke A, Engler RJM, Nelson MR, Collins LC, Spooner C, Guthridge C, James JA. Insufficient Anthrax Lethal Toxin Neutralization Is Associated with Antibody Subclass and Domain Specificity in the Plasma of Anthrax-Vaccinated Individuals. Microorganisms. 2021 Jun 2;9(6):1204. doi: 10.3390/microorganisms9061204. PMID: 34199431; PMCID: PMC8229884.

Bower WA, Schiffer J, Atmar RL, Keitel WA, Friedlander AM, Liu L, Yu Y, Stephens DS, Quinn CP, Hendricks K; ACIP Anthrax Vaccine Work Group. Use of Anthrax Vaccine in the United States: Recommendations of the Advisory Committee on Immunization Practices, 2019. MMWR Recomm Rep. 2019 Dec 13;68(4):1-14. doi: 10.15585/mmwr.rr6804a1. PMID: 31834290; PMCID: PMC6918956.

5) '9. Persistence of PA in the blood' The paragraph on TPE should be removed, it is unclear why it is present.  If you have to remove your vaccine antigen from the blood after vaccination, there is a serious problem.  Even if the assertion that PA in the blood is somehow responsible for GWI is true, there is no PA in the blood decades after vaccination at the time the symptoms begin.  Further, if the assertion is true, it would be trivial to determine what parts of PA63 are responsible for the toxicity in the authors assays and engineer a vaccine with mutated/truncated PA as should have been detailed in the subunit vaccine section.

6) '10. future vaccines' The authors tout the discussion about an mRNA vaccine in the abstract, yet it is relegated to a tiny paragraph in this review.  Ironically, the COVID-19 vaccine is also showing IgG4 subclass preferences after boosts indicating that an mRNA vaccine route resulting in persistent, long-lived antigen production will have the exact same issues detailed in this review, and maybe even worse than AVA.

7)  Sections 11 and 12  The assertion that any vaccine will require HLA-sequencing of each vaccinee to see if the vaccine is appropriate and safe is simply absurd.  

Author Response

Comments and Suggestions for Authors

This review is very well written and sections 1-5 are very good.

Thank you!

Title: Unfortunately, the current title was used by Turnbull in 1991! So we changed the title to “Anthrax vaccines in the 21st century”.

Turnbull PC. Anthrax vaccines: past, present and future. Vaccine. 1991 Aug;9(8):533-9. doi: 10.1016/0264-410x(91)90237-z.

Major criticisms: 

1)  '6. Safety' The second paragraph assumes and essentially asserts that the anthrax vaccine is the cause of GWI and because of this, the anthrax vaccine will need "rigorous assessment of various organ systems over months and years is necessary".  Such assessments will never be accomplished in a modern vaccine trial and it is ironic that this review touts rushed mRNA vaccine technology - the exact opposite of this point.

REPLY: Made it more concise and to the point.

2)  '7. Neurotoxicity...'  This entire paragraph should be reduced to 2-3 sentences and folded into the previous section or section 9.  Readers looking for a review on the anthrax vaccine likely have little interest in the in vitro neurotoxicity of PA63.

REPLY: We did.

3) '8.3. PA-free vaccines' I don't really understand this paragraph at all, with regard to 15-aa epitopes.  This paragraph should be about subunit vaccines, much work has gone into the feasibility of a domain 4 vaccine.  Various possibilities of using PA without PA-20 and domain 3 and PA-domain/LF chimeras should be discussed.  Furthermore, the assertion that vaccine trials need to determine the vaccine's potential for inducing autoimmunity is untenable.  What would those experiments even be?  It would take decades for proper human data.

REPLY: We revised accordingly.

4) '9. Persistence of PA in the blood' A paragraph needs to be added about the immunologic features of AVA and persistent exposure to the vaccinating antigen due to slow release and frequent boosts.  Several studies have shown that the AVA schedule causes induction of IgG4 subclass anti-PA which may have a detrimental effect on neutralization capacity.  The recent ACIP update has somewhat addressed this (citations below).  This is very applicable to this review.

Quinn CP, Sabourin CL, Schiffer JM, Niemuth NA, Semenova VA, Li H, Rudge TL, Brys AM, Mittler RS, Ibegbu CC, Wrammert J, Ahmed R, Parker SD, Babcock J, Keitel W, Poland GA, Keyserling HL, El Sahly H, Jacobson RM, Marano N, Plikaytis BD, Wright JG. Humoral and Cell-Mediated Immune Responses to Alternate Booster Schedules of Anthrax Vaccine Adsorbed in Humans. Clin Vaccine Immunol. 2016 Apr 4;23(4):326-38. doi: 10.1128/CVI.00696-15. PMID: 26865594; PMCID: PMC4820509.

Smith K, Garman L, Norris K, Muther J, Duke A, Engler RJM, Nelson MR, Collins LC, Spooner C, Guthridge C, James JA. Insufficient Anthrax Lethal Toxin Neutralization Is Associated with Antibody Subclass and Domain Specificity in the Plasma of Anthrax-Vaccinated Individuals. Microorganisms. 2021 Jun 2;9(6):1204. doi: 10.3390/microorganisms9061204. PMID: 34199431; PMCID: PMC8229884.

Bower WA, Schiffer J, Atmar RL, Keitel WA, Friedlander AM, Liu L, Yu Y, Stephens DS, Quinn CP, Hendricks K; ACIP Anthrax Vaccine Work Group. Use of Anthrax Vaccine in the United States: Recommendations of the Advisory Committee on Immunization Practices, 2019. MMWR Recomm Rep. 2019 Dec 13;68(4):1-14. doi: 10.15585/mmwr.rr6804a1. PMID: 31834290; PMCID: PMC6918956.

REPLY: Absolutely! Thank you for pointing this out. We have now discussed these issues in the revised paper.

5) '9. Persistence of PA in the blood' The paragraph on TPE should be removed, it is unclear why it is present.  If you have to remove your vaccine antigen from the blood after vaccination, there is a serious problem.  Even if the assertion that PA in the blood is somehow responsible for GWI is true, there is no PA in the blood decades after vaccination at the time the symptoms begin.  Further, if the assertion is true, it would be trivial to determine what parts of PA63 are responsible for the toxicity in the authors assays and engineer a vaccine with mutated/truncated PA as should have been detailed in the subunit vaccine section.

REPLY: Removed.

6) '10. future vaccines' The authors tout the discussion about an mRNA vaccine in the abstract, yet it is relegated to a tiny paragraph in this review.  Ironically, the COVID-19 vaccine is also showing IgG4 subclass preferences after boosts indicating that an mRNA vaccine route resulting in persistent, long-lived antigen production will have the exact same issues detailed in this review, and maybe even worse than AVA.

REPLY: Very true! We are discussing this issue now in the revised paper.

7)  Sections 11 and 12  The assertion that any vaccine will require HLA-sequencing of each vaccinee to see if the vaccine is appropriate and safe is simply absurd. 

REPLY: Removed.

Submission Date

12 December 2023

Date of this review

03 Jan 2024 23:06:01

© 1996-2024 MDPI (Basel, Switzerland) unless otherwise stated

Reviewer 2 Report

Comments and Suggestions for Authors

The review “Anthrax vaccine: Past, present, and future” is a paper describing the main anthrax vaccines available for human use in the past and in the present, with particular reference to safety and neurotoxicity to PA. Furthermore, the review also describes new anthrax vaccines under study and new possible vaccines that could be developed such as a mRNA-based.

The review is interesting and well discussed with good insight about the host immunogenetics and the effective immune response.

Nevertheless, it contains some points to be clarified and integrated before the paper may become publishable.

Title: I would use “anthrax vaccines: past, present and future” since you speak about different vaccines.

Line 59: detail the strain used for AVA vaccine.

Lines 73-75: Describe where and when this vaccine has been developed.

Paragraph 4: It would be useful to provide a graphic or a table with the antibody production of the different vaccines.

Paragraph 6 “Safety”: In this paragraph is not clearly specified about which vaccine you are speaking. Please clarify better for the reader. Please write the extended name of GWI al the least in the first time you write about it.

Line 179: What do you mean with “undefined composition” and “lot to lot variation”? please explain better.

Paragraph 8.2: I think they are experimental vaccines, so I would call the paragraph “Candidate other PA-based vaccines”.

Paragraph 8.3: Since we are speaking about vaccines made with epitopes of PA (so parts of PA), maybe the title of this paragraph could be misleading.

Line 211: Replace “and” with “an”.

Line 209: Delete one space before “A vaccine”.

Line 221: Delete one space before “Although”.

Lines 335-343: It’s not clear this part. It’s not well clear what you want to say in correlation to protection. Please explain better.

Line 360 Correct “vaccines”.

Author Response

Comments and Suggestions for Authors

The review “Anthrax vaccine: Past, present, and future” is a paper describing the main anthrax vaccines available for human use in the past and in the present, with particular reference to safety and neurotoxicity to PA. Furthermore, the review also describes new anthrax vaccines under study and new possible vaccines that could be developed such as a mRNA-based.

The review is interesting and well discussed with good insight about the host immunogenetics and the effective immune response.

Nevertheless, it contains some points to be clarified and integrated before the paper may become publishable.

Title: I would use “anthrax vaccines: past, present and future” since you speak about different vaccines.

REPLY: Unfortunately, this title was used by Turnbull in 1991! So we changed the title to “Anthrax vaccines in the 21st century”.

Turnbull PC. Anthrax vaccines: past, present and future. Vaccine. 1991 Aug;9(8):533-9. doi: 10.1016/0264-410x(91)90237-z.

Line 59: detail the strain used for AVA vaccine.

REPLY: Unfortunately, the strain of B. anthracis used for the current AVA vaccine is not stated in official websites, namely the:

BioThrax manufacturer website (https://www.emergentbiosolutions.com/products-services/our-products “BioThrax is manufactured from a culture filtrate, made from a non-virulent strain of Bacillus anthracis”)

CDC website (https://www.cdc.gov/vaccines/vpd/anthrax/hcp/about-vaccine.html “Each 0.5 milliliter (mL) dose of anthrax vaccine adsorbed or BioThrax® (Emergent BioSolutions) is made from cell-free filtrates of microaerophilic cultures of an avirulent, nonencapsulated strain of Bacillus anthracis

The classic citation on the original production (Puziss M, Manning LC, Lynch JW, Barclaye, Abelow I, Wright GG. Large-scale production of protective antigen of Bacillus anthracis in anaerobic cultures. Appl Microbiol. 1963 Jul;11(4):330-4. doi: 10.1128/am.11.4.330-334.1963.) mentions that “Strain V770-NPl-R, a nonencapsulated, nonproteolytic, and avirulent mutant of B. anthracis was used”. We could not find any information that this is the strain used for production of the current AVA BioThrax vaccine. We discuss this issue in the revised paper.

Lines 73-75: Describe where and when this vaccine has been developed.

REPLY: Done and placed in Section 8 since it is one of the newer vaccines.

Paragraph 4: It would be useful to provide a graphic or a table with the antibody production of the different vaccines.

REPLY: Unfortunately, this is not practically feasible, given the diversity of antibodies (e.g. different subclasses of IgG), the different immunization schedules, and other performance and timing of other tests, in addition to antibody titers (e.g. TNA, IFN-γ, IL-4, etc.). We believe that the essential information on this topic is provided in the review.

Paragraph 6 “Safety”: In this paragraph is not clearly specified about which vaccine you are speaking. Please clarify better for the reader. Please write the extended name of GWI al the least in the first time you write about it.

REPLY: Done for both points.

Line 179: What do you mean with “undefined composition” and “lot to lot variation”? please explain better.

REPLY: We have added the relevant reference (Friedlander AM, Little SF. Advances in the development of next-generation anthrax vaccines. Vaccine. 2009 Nov 5;27 Suppl 4:D28-32. doi: 10.1016/j.vaccine.2009.)

Paragraph 8.2: I think they are experimental vaccines, so I would call the paragraph “Candidate other PA-based vaccines”.

REPLY: Changed.

Paragraph 8.3: Since we are speaking about vaccines made with epitopes of PA (so parts of PA), maybe the title of this paragraph could be misleading.

REPLY: Correct. Changed to PA63 subunit vaccines.

Line 211: Replace “and” with “an”.

REPLY: Done.

Line 209: Delete one space before “A vaccine”.

REPLY: Done.

Line 221: Delete one space before “Although”.

REPLY: Done.

Lines 335-343: It’s not clear this part. It’s not well clear what you want to say in correlation to protection. Please explain better.

REPLY: We did.

Line 360 Correct “vaccines”.

REPLY: Actually, we mean “vaccinees”, i.e. persons to be vaccinated.

 Submission Date

12 December 2023

Date of this review

29 Dec 2023 20:27:03

© 1996-2024 MDPI (Basel, Switzerland) unless otherwise stated

Reviewer 3 Report

Comments and Suggestions for Authors

The proposed manuscript highlights some of the key issues in the anthrax vaccine space. However, there were multiple areas of the manuscript that would benefit from additional information and/or context noted below.

Major comments:

·       Lines 53-56: This is a bit of an incomplete summary of anthrax vaccine development efforts, both current and from the recent past. It may be worth emphasizing the point that many of the most recent R&D investments had focused on PA-only based vaccines.

·       Line 70: The statement around AVA administration (IM) is true for pre-exposure prophylaxis but the post-exposure regimen is by subcutaneous administration. The route for CYFENDUS should be included.

·       Section 4: While I agree with the complications around using immunological readouts to predict protective efficacy, this section should more clearly note that from a regulatory perspective, this is likely the only pathway to licensure absent large numbers of anthrax cases that would warrant traditional phase 3 clinical trials. And it is the toxin neutralization assay, specifically measure by its NF50 value, that has been used for licensure in the US.

·       Section 5: With CYFENDUS being approved in the US, it should probably be added to this section.

·       Lines 111-113: This sentence may benefit from an expansion to include notes on rates of AE’s and specifically Grade 3 and above.

·       Line 189: While the intent was understood, readers may be unclear about AVA composition by saying it is PA only.

·       Line 196: The authors should provide references for any indication of reduced or eliminated lot-lot variation among PA based vaccines. Unless a reference can be provided related to a validated manufacturing process, that may be an over statement.

·       Section 10: This section would require additional context in terms of pros and cons. For example, the idea that COVID vaccines have paved the way for mRNA based anthrax vaccines is an over-simplification. For example, a viral antigen will ultimately be produced in mammalian cells during infection and the same is true in an mRNA based vaccine. It is not clear if mammalian expression of PA would elicit protective immune responses. mRNA based vaccines require storage under frozen conditions and it is unclear if such vaccines would be stable long term.

·       Line 342: While the sentiment behind the notion that any cases of anthrax is too many, it may be unrealistic to expect a vaccine with efficacy higher than the estimated 92.5% cited.

Author Response

Comments and Suggestions for Authors

The proposed manuscript highlights some of the key issues in the anthrax vaccine space. However, there were multiple areas of the manuscript that would benefit from additional information and/or context noted below.

Thank you!

Title: Unfortunately our title was used by Turnbull in 1991! So we changed the title to “Anthrax vaccines in the 21st century”.

Turnbull PC. Anthrax vaccines: past, present and future. Vaccine. 1991 Aug;9(8):533-9. doi: 10.1016/0264-410x(91)90237-z.

Major comments:

  • Lines 53-56: This is a bit of an incomplete summary of anthrax vaccine development efforts, both current and from the recent past. It may be worth emphasizing the point that many of the most recent R&D investments had focused on PA-only based vaccines.

REPLY: We expanded this paragraph with more details.

  • Line 70: The statement around AVA administration (IM) is true for pre-exposure prophylaxis but the post-exposure regimen is by subcutaneous administration. The route for CYFENDUS should be included.

REPLY: Done.

  • Section 4: While I agree with the complications around using immunological readouts to predict protective efficacy, this section should more clearly note that from a regulatory perspective, this is likely the only pathway to licensure absent large numbers of anthrax cases that would warrant traditional phase 3 clinical trials. And it is the toxin neutralization assay, specifically measure by its NF50 value, that has been used for licensure in the US.

REPLY: We agree, and revised accordingly.

  • Section 5: With CYFENDUS being approved in the US, it should probably be added to this section.

REPLY: Added.

  • Lines 111-113: This sentence may benefit from an expansion to include notes on rates of AE’s and specifically Grade 3 and above.

REPLY: We did.

  • Line 189: While the intent was understood, readers may be unclear about AVA composition by saying it is PA only.

REPLY: We clarified this point better.

  • Line 196: The authors should provide references for any indication of reduced or eliminated lot-lot variation among PA based vaccines. Unless a reference can be provided related to a validated manufacturing process, that may be an over statement.

REPLY: We did.

  • Section 10: This section would require additional context in terms of pros and cons. For example, the idea that COVID vaccines have paved the way for mRNA based anthrax vaccines is an over-simplification. For example, a viral antigen will ultimately be produced in mammalian cells during infection and the same is true in an mRNA based vaccine. It is not clear if mammalian expression of PA would elicit protective immune responses. mRNA based vaccines require storage under frozen conditions and it is unclear if such vaccines would be stable long term.

REPLY: We agree, and revised accordingly.

  • Line 342: While the sentiment behind the notion that any cases of anthrax is too many, it may be unrealistic to expect a vaccine with efficacy higher than the estimated 92.5% cited.

REPLY: We agree, and revised the statement accordingly.

Submission Date

12 December 2023

Date of this review

26 Dec 2023 21:54:54

© 1996-2024 MDPI (Basel, Switzerland) unless otherwise stated

Disclaimer

Round 2

Reviewer 1 Report

Comments and Suggestions for Authors

The authors did an excellent job addressing all of my comments and the changes are satisfactory for publication.

Author Response

Thank you!

Reviewer 2 Report

Comments and Suggestions for Authors

Thank you for providing the revised version. The paper has been significantly modified and improved.

I noticed problems about references. There are some bibliographic entries that are not present in the text (from n. 78 to n. 84).

Line 289: Replace the square bracket with the round bracket

Line 376: Is it a mistake the exclamation mark?

Line 403: The reference is missing

Author Response

We thank the reviewer for their efforts and very useful suggestions that improved substantially the paper!

In this 2nd revision, we corrected all issues mentioned by the reviewer.

Reviewer 3 Report

Comments and Suggestions for Authors

While the attempts to revise based on reviewer comments was appreciated, the responses were incomplete in some cases and resulted in additional questions/issues in others. The second review would have been helped if the authors provided point-by-point responses instead of 'we did' in most cases. This manuscript may be better served by focusing the content. By that, I mean that the authors seem to be going down two different avenues, one being the need for R&D in a next-generation of anthrax vaccines and the other being potential long-term safety concerns with the existing vaccines. While the two are clearly inter-related topics, the authors may have benefited by focusing on one or the other in that neither focus is fully addressed in the current version.

Author Response

While the attempts to revise based on reviewer comments was appreciated, the responses were incomplete in some cases and resulted in additional questions/issues in others. The second review would have been helped if the authors provided point-by-point responses instead of 'we did' in most cases.

REPLY: The revision was fairly extensive and included rearrangement of sections and text, so it was practically impossible to pinpoint the exact place where the specific correction/revision was made. We are sorry for the confusion this caused.

This manuscript may be better served by focusing the content. By that, I mean that the authors seem to be going down two different avenues, one being the need for R&D in a next-generation of anthrax vaccines and the other being potential long-term safety concerns with the existing vaccines. While the two are clearly inter-related topics, the authors may have benefited by focusing on one or the other in that neither focus is fully addressed in the current version.

REPLY: We agree that 2 separate reviews would certainly cover the vast materials better but at the price, we believe, that the forest would have been lost to the trees, if we may use that expression. We believe that we have outlined the various issues clearly and provided good summaries and representative references to cover the intended topic and, at the same time, inspire other researchers to amplify individual topics.